# Impact of Diabetes Mellitus on Bone Health

**DOI:** 10.3390/ijms20194873

**Published:** 2019-09-30

**Authors:** Cliodhna E. Murray, Cynthia M. Coleman

**Affiliations:** Regenerative Medicine Institute, National University of Ireland, Galway, Biomedical Sciences Building, Dangan, Newcastle Road, Galway City, County Galway H91W2TY, Ireland; cynthia.coleman@nuigalway.ie

**Keywords:** diabetes mellitus, type 1 complications, type 2 complications, bone remodeling, fracture healing, bone marrow dysfunction, mesenchymal stem cells

## Abstract

Long-term exposure to a diabetic environment leads to changes in bone metabolism and impaired bone micro-architecture through a variety of mechanisms on molecular and structural levels. These changes predispose the bone to an increased fracture risk and impaired osseus healing. In a clinical practice, adequate control of diabetes mellitus is essential for preventing detrimental effects on bone health. Alternative fracture risk assessment tools may be needed to accurately determine fracture risk in patients living with diabetes mellitus. Currently, there is no conclusive model explaining the mechanism of action of diabetes mellitus on bone health, particularly in view of progenitor cells. In this review, the best available literature on the impact of diabetes mellitus on bone health in vitro and in vivo is summarised with an emphasis on future translational research opportunities in this field.

## 1. Introduction

Impaired bone quality and increased fracture risk have become recognized complications of diabetes mellitus [1]. Two meta-analyses involving a total of 7,832,213 participants found an increased incidence of hip fractures in individuals living with diabetes mellitus compared to the general population, whereby those living with type 1 diabetes mellitus (T1DM) (relative risk (RR)= 5.76–6.3) show a higher incidence than individuals living with type 2 diabetes mellitus (T2DM) (RR = 1.34–1.7) [2,3]. In addition, diabetic fracture risk benefits significantly from effective clinical management, as fracture risk is higher in diabetes mellitus with poor glycemic control compared to adequately controlled diabetes mellitus [4,5]. 

The increased fracture risk in individuals living with diabetes mellitus is compounded by impaired fracture healing. Specifically, alterations in bone metabolism and the development of microvascular disease can prolong healing time by 87% [6]. Additionally, patients living with diabetes mellitus are predisposed to an increased risk of complications such as delayed wound closure [7], infectious complications [8], and peri-operative cardiovascular events [9]. Considering the higher incidence of diabetes mellitus and the considerable socioeconomic burden generated by fragility fractures [10], these findings draw attention to the need for an improved awareness of the factors that determine bone health and the risk of fracture in patients living with diabetes mellitus. The aim of this narrative review is to summarise the best available topical literature in order to create a better understanding of the interaction of bone health and diabetes mellitus on a molecular level, and to draw attention to future areas of research in this field. To achieve this aim, publications containing the terms “bone AND diabetes” were evaluated using PubMed Central. The search was limited to title or abstract between the years 2000 and 2019. Reference lists of the identified publications were evaluated to identify additional relevant studies.

## 2. Bone Mineral Density

Patients living with T1DM are affected by a complete failure of β-cells of the pancreas combined with low levels of insulin-like growth factor 1 (IGF1). Both the lack of insulin, among other pancreatic anabolic hormones, and low IGF1 levels suppress the terminal differentiation of mesenchymal stem cells (MSCs) into osteoblasts in addition to osteoblastic activity [11]. Therefore, this inhibits skeletal growth at a young age, which leads to an inadequate accrual of peak bone mass [12,13,14,15,16]. On the contrary, T2DM affects bone health in advanced stages of the disease where many factors such as insulinopenia, hyperglycemia, the development of advanced glycation end products (AGEs), chronic inflammation, and microvascular disease coincide to negatively affect bone architecture and biomechanical properties of the bone (Figure 1) [17,18]. As a result, the relative risk of sustaining a hip fracture increases over the course of T2DM [1].

Whereas T1DM is associated with modest reductions in bone mineral density (BMD) (Hip Z-scores of −0.37  ±  0.16) [19] and an increase in fracture risk [20], patients living with T2DM have higher BMD (Hip Z-scores of 0.27 ± 0.16 [19]) with an increased fracture risk [19,21,22]. This contradiction can be explained as follows. Individuals living with diabetes mellitus suffer from a higher incidence of falls due to long-term complications. However, in a meta-analysis, after factoring out for increased falls as well as other confounders, such as hypoglycemic episodes and the use of anti-diabetic medications, patients with T2DM still had an increased risk of a fracture [23,24]. Therefore, the literature suggests that there is independence of fracture risk in diabetes mellitus to both changes in BMD and increased risk of falls. This can be explained by impairments of bone architecture [24].

The investigation of bone architecture in individuals living with diabetes mellitus has been facilitated by the development of non-invasive imaging techniques [25,26]. A study using high-resolution peripheral quantitative computer tomography shows T2DM is associated with a 10% higher trabecular BMD and an increase in intracortical porosity [27]. Some recent imaging studies suggest higher adiposity and an increased fraction of saturated fat in the bone marrow of patients living with diabetes mellitus. However, so far, these studies have not adjusted for obesity-related bone marrow adiposity [28,29]. Recently, changes in bone structure were directly confirmed using in vivo micro-indentation of the tibia to measure bone micro-architecture in patients with T2DM compared to the controls. These patients showed significantly increased cortical porosity and a significantly lower bone mineral strength than healthy controls [30].

## 3. Biochemical Impact on Bone Micro-Architecture 

Extracellular bone matrix is composed of two materials. The inorganic mineral component, consisting mainly of hydroxyapatite, provides stiffness, which is the quality that is measured by a conventional BMD scan. The organic component, composed predominantly of interconnecting collagen fibers [31], provides tensile strength and counteracts shear stresses [32]. These material properties of bone tissue are regulated by cellular activity, bone tissue turnover rate, and collagen cross-link formation [32,33]. Meanwhile, these cellular activities are influenced by many environmental factors, including circulating hormones, oxidative stress, and level of glycation [34,35,36], as summarised in Figure 1.

Indirectly, many additional factors associated with hyperglycemia affect bone micro-architecture in diabetes mellitus. For example, glycosuria proportionally increases calcium excretion in urine [37]. Additionally, the interaction of hyperglycemia with the parathyroid hormone (PTH) and vitamin D system affects bone turnover in the population of patients living with diabetes mellitus (Figure 1) [34,38]. One meta-analysis in 2007 found evidence that vitamin D and calcium supplementation may be important for preventing T2DM in patients with impaired glucose tolerance [39].

### 3.1. Insulin Signalling

The literature suggests that insulin, as well as other pancreatic hormones, serve as anabolic factors in bone formation [34,40]. In one in vitro study, conditional disruption of the gene encoding for the IGF1 receptor in osteoblasts negatively impacts their proliferation and mineralisation. However, this defect was rescued by insulin treatment. Additionally, in vivo evidence in a murine model suggests that IGF1 plays a central role in the terminal differentiation of MSCs into osteoblasts [11]. Therefore, insulin exerts direct action in the regulation of osteoblastic activity by activation of its cell surface receptor, and IGF1 modulates the strength of the insulin-generated signal through interactions with the IGF1 receptor (Figure 1) [40]. In T1DM, absolute insulinopenia in combination with low levels/low action of IGF1 decrease bone formation by exerting an inhibitory effect on osteoblasts and their progenitor cells in the early stages of the disease [17]. However, in T2DM, this inhibitory effect caused by insulinopenia and low levels of IGF1 would be expected in advanced stages of the disease [17]. Since T1DM typically occurs in children, adolescents, and young adults, the state of absolute insulinopenia corresponds with a stage of skeletal maturation. Therefore, these studies suggest that particularly inadequately controlled T1DM will impact bone accrual and the development of peak bone mass [34]. 

### 3.2. Hyperglycemia and AGEs

A hyperglycemic environment exerts a direct and indirect effect on the function and differentiation of osteoblasts [41,42]. In vitro studies show hyperglycemia directly affects the metabolism and maturation of osteoblasts by altering gene expression [41,42] and, thereby, diminishing the quality of the bone mineral [43]. Additionally, it has been demonstrated that hyperglycemia increases the level of pro-inflammatory cytokines in humans, such as tumor necrosis factor alpha (TNF-α), interleukin 1 beta, interleukin 6, interleukin 8, and interleukin 8 [43,44] while simultaneously increasing the receptor activator of nuclear factor kappa-Β ligand (RANKL) expression [43], which mediates osteoblast death and osteoclastogenesis, respectively (Figure 1) [35]. Since inflammatory factors are elevated in the early stages of T1DM [45], the above named pro-inflammatory cytokines could play a role in the inhibited accrual of bone mass [46].

The evidence shows that oxidative stress and a hyperglycemic metabolic state, which are induced and maintained by diabetes mellitus, lead to the accelerated formation of AGEs (for example, pentosidine) [35,47,48,49]. AGEs cross-link with collagen fibers in both trabecular and cortical bone [50], which leads to a more brittle bone with a deterioration of post-yield properties (making bones less able to deform before fracturing) (Figure 1) [36,51]. In contrast, physiological enzymatic cross-links between collagen fibers provide a beneficial effect on the quality and strength of the bone [36]. In spontaneously diabetic WBN/Kob rats, a steady decrease of beneficial enzymatic cross-links coupled with a steady increase of pentosidine was reported after onset of diabetes mellitus. Additionally, impaired bone biomechanics coincided with these alterations in collagen cross-linking, despite no alterations in BMD values [52]. Therefore, AGEs are thought to deteriorate biomechanical function of the bone by altering the physical properties of bone collagen, which results in bone fragility [53].

Accompanying the alteration in collagen cross-links, AGEs affect bone tissue by directly interfering with the development [54] and function [55] of bone cells. AGEs affect the phenotypic expression of osteoblasts in vitro, in particular inhibiting nodule formation of osteoblasts in a cell culture [56]. In addition, AGEs may decrease bone resorption by inhibiting osteoclastic differentiation activity and, thereby, altering the structural integrity of the collagen matrix [57]. It has been established that the osteoblastic function is disrupted by AGEs by upregulating the cell surface receptor for advanced glycation end products (RAGE) located on osteoblasts (Figure 1) [58,59]. These receptors have been shown to increase the production of pro-inflammatory cytokines, which may feed a cycle of increased bone resorption and chronic inflammation [60]. Furthermore, one study shows that treating an osteocytic cell line with AGEs increases sclerostin expression and decreases RANKL expression. Therefore, this suppresses bone formation and bone resorption, respectively [61]. These adverse effects of AGEs on bone cells serve to further accelerate bone fragility in diabetes mellitus [33]. Galectin-3 protein in bone tissue has been shown to play an important role in the uptake and removal of AGEs whereby Galectin-3 exerts the opposite action on AGE-receptor to RAGE. Therefore, this potentially serves as a protective factor in diabetes mellitus-related AGE accumulation [62,63]. 

## 4. Epigenetic Changes 

Large clinical trials have shown that diabetic complications in T1DM and T2DM continue to progress after patients return to adequate glycemic control [64,65,66,67,68,69]. Additionally, it is known that HbA1c merely accounts for 25% of the variation in the risk of developing complications, which implies that transient hyperglycemic episodes lead to lasting cellular changes [66,70]. Recent investigations, particularly in murine models of cardiovascular disease, have begun to shed light on the patho-mechanism of metabolic memory in diabetes mellitus, which leads to the development of end-organ damage [64,71,72,73,74,75]. For instance, microRNA (miRNA)-155 was decreased in streptozotocin-induced diabetic rats and negatively correlated to NF-κB activity and an apoptosis rate [76]. This was reflected in a study showing a downregulation of miRNA-155 in bone-marrow derived progenitor cells isolated from humans living with T2DM [77]. In a clinical study, gene expression of p66Shc in peripheral mononuclear cells was correlated with new onset complications in patients living with diabetes mellitus with similar baseline characteristics [78]. These recent findings draw attention to the importance of early and aggressive treatment of uncontrolled diabetes mellitus. Uncovering epigenetic therapeutic targets will open opportunities for the development of drugs to improve patients’ outcome after glucose homeostasis has been achieved [65,79,80].

## 5. Bone Turnover

The effect of a diabetic environment on bone metabolism can be indirectly measured through bone turnover markers. Specifically, osteocalcin is produced by osteoblasts and is a marker of bone formation [81]. Children suffering from T1DM were found to have low levels of osteocalcin, which were negatively correlated with HbA1c levels [82,83]. Derivatives of furanocoumarins reversed the suppression of osteocalcin and diabetes mellitus-associated decreased trabecular thickness in diabetic mice, in addition to significantly suppressing osteoclast-related gene expression such as RANKL [84]. When comparing T1DM and T2DM, osteocalcin serum levels are decreased in individuals living with T1DM and significantly decreased in T2DM compared to healthy controls [82,85,86,87]. Alternatively, sclerostin is a marker for bone resorption [81] and is inversely correlated to bone turnover markers for bone formation in patients living with T2DM [88,89,90]. However, changes in sclerostin levels have not been confirmed for individuals living with T1DM [88]. Bone turnover markers could potentially be a means of predicting the fracture risk in patients living with diabetes mellitus in the future [91,92,93]. 

“Signature miRNAs” of bone turnover, such as miR-148a-3p, are known as biomarkers in primary osteoporosis [94,95,96]. In 2016, Heilmeyer et al. studied circulating miRNAs and identified combinations of miR-550a-5p, miR-96-5p, miR-382-3p, and miR-181c-5p associated with T2DM-induced fragility fractures with a high specificity and sensitivity [97]. This study also included an in vitro analysis to measure the effect of miR-550a-5p, miR-382-3p, and miR-188-3p on adipose tissue-derived MSCs. Interestingly, miR-382-3p was found to stimulate osteogenic differentiation and inhibit adipogenesis. This could be explained by the fact that the level of miR-382-3p was seven times lower in fractured patients living with T2DM compared to T2DM without a history of fragility fractures. On the contrary, miR-550a-5p was upregulated 22-fold in the diabetes fracture group compared to non-fracturing patients living with T2DM, and was shown to be a strong inhibitor of osteogenesis [97]. In T1DM, hyperexpression of miR-148a and miR-21-5p was observed in the sera of patients, which was associated with decreased BMD and increased circulating PTH levels [98]. 

Studies examining the effect of diabetes mellitus on osteoclasts are not conclusive. In vitro and animal studies report an unaltered rate of bone resorption [99,100], whereas some studies have suggested increased osteoclastic activity in diabetes mellitus under certain conditions, such as periodontal disease [101] and osteoporosis [102]. Other studies have even reported inhibited osteoclast function and differentiation in a diabetic environment [103,104,105]. Due to the conflicting evidence and generally negligent effect that has been observed in osteoclasts, it seems likely that the impaired bone formation in diabetes mellitus is primarily due to inhibited osteoblastic and progenitor cell activity rather than an alteration of bone resorption. However, further research is needed to clarify the effect of diabetes mellitus on osteoclastic function and differentiation. 

## 6. Fracture Risk

Altered biomechanical properties of the bone due to deteriorations in bone microarchitecture predispose individuals living with diabetes mellitus to fragility fractures [106,107,108]. Individuals living with T2DM and T1DM carry a higher risk of sustaining a fracture at most skeletal locations compared to the general population, whereby hip fractures in T2DM has been most extensively examined [109,110,111]. T1DM is reported to be associated with a higher odds ratio for hip fractures compared to hip fractures in patients living with T2DM in a meta-analysis [19]. When fractures are compared by anatomical location in T2DM, women living with diabetes mellitus have a significantly increased risk of hip, pelvis, upper leg, foot, and vertebral fractures [112]. Additionally, diabetes mellitus is a negative prognostic factor for post-fracture mortality among patients with hip fractures [17,113,114]. However, despite the increased fracture risk, patients with T2DM show a higher BMD at the femoral neck and lumbar spine in conventional Dual-energy X-ray absorptiometry (DEXA) scans [115].

Accumulation of AGEs, specifically pentosidine, is associated with a fracture incidence in older adults living with diabetes mellitus, as demonstrated by Schwartz et al. in the Health Aging and Body Composition study [116]. Similarly, a high level of urinary excretion of pentosidine in non-diabetic patients was an independent risk factor for vertebral fractures [117]. One clinical study shows increased cortical bone AGEs in T2DM patients [118]. Additionally, another study reports that trabecular bone from fracturing T1DM patients has significantly higher levels of pentosidine than non-fracturing T1DM [119], even though this does not imply causality. Large retrospective studies have shown that conventional models for predicting fracture risk such as BMD and the Fracture Risk Assessment Tool (FRAX) underestimate the fracture risk for patients living with diabetes mellitus due to secondary impairments in bone micro-architecture [120,121]. However, the trabecular bone score, which is related to the bone micro-architecture, was shown to predict fractures in patients suffering from diabetes mellitus with greater accuracy [122,123,124]. 

## 7. Fracture Healing

In usual fracture healing, a stabilising callus is formed, in which cartilage is formed and then reabsorbed and replaced by bone tissue. This is facilitated by blood supply to the healing site [125]. In animal models of fracture healing, many studies have suggested diabetes mellitus is associated with an impaired healing response [126,127,128,129,130]. In a diabetic murine model, the animals were shown to have an increased concentration of TNF-α at the fracture site, which was linked to an increased rate of cartilage resorption [127]. Additionally, a diabetic cell environment may lead to a reduction in callus size and bone formation and, thereby, a decrease in the mechanical strength of the repaired fracture site [126,127,128]. In one in vivo study, decreased cell proliferation as well as decreased mechanical stiffness was shown at the fracture site of poorly controlled diabetic rats. However, rats with a tight insulin treatment maintained physiological fracture healing [131]. In healthy human individuals, there is a fracture response during the first few weeks of recovery marked by a peak in osteocalcin, alkaline phosphatase (ALP), and IGF1, which indicates increased bone turnover [132,133]. However, in individuals living with diabetes mellitus, bone turnover markers post-fracture are diminished [134], which could possibly be a symptom of disturbed fracture consolidation. 

Fracture healing is intimately associated with progenitor cell population and functionality [135,136]. One study demonstrates atrophic non-union fractures are associated with a decreased pool of MSCs, which alters the level of chemokines involved in fracture healing [137]. Therefore, insufficient MSC availability may impede callus remodeling and result in callus material that is biomechanically inferior in patients living with diabetes mellitus [130,138,139,140]. Long-term complications of patients living with diabetes mellitus include microvascular complications [141], where complications such as fracture non-union are linked to vascular insufficiencies in the fracture site [142,143]. Since vascularization is mediated by MSCs [144,145], vascular deficiencies may be further impaired in diabetic fracture healing due to the reduced population and potential of progenitor cells and chronic inflammatory environment. Several studies have shown a decreased expression of angiogenic genes (VEGF-A, VEGF-C, angiopoietin 1, and angiopoietin 2) and proteins in MSCs isolated from humans living with diabetes mellitus [146,147]. In addition to these impediments, patients living with diabetes mellitus have a greater risk of wound infection, local post-operative complications such as impaired wound healing, and peri-operative cardiovascular complications compared to non-diabetic individuals [6,8,9,148].

## 8. Effect of Diabetes on Progenitor Cells 

Adipocytes and osteoblasts are derived from a common precursor, the MSC. The differentiation of MSC is influenced by the interaction of several different pathways (Figure 1). The WNT signaling and peroxisome proliferator-activated receptor gamma (PPAR-*γ*) pathways regulate a fine balance between adipogenesis and osteo-blastogenesis [149]. The activation of the WNT signaling pathway promotes osteogenesis and inhibits adipogenesis. On the contrary, PPAR-*γ*, which is mediated by reactive oxygen species (ROS) [150], facilitates the differentiation of MSCs into adipocytes [18]. In one study, muscle-derived MSCs cultured in high glucose media showed a higher expression of adipogenesis markers (PPAR-*γ*, LPL, adiponectin, GLUT4, and SREBP1c) and a down-regulation of chondrogenic and osteogenic markers compared to cells cultured in a low glucose media [150]. In a similar model, gene expression associated with osteoblast differentiation was decreased, with a simultaneous increase in cells of an adipocyte phenotype in a hyperglycemic environment [151]. A recent study utilising rat bone-marrow derived (BM-)MSCs has suggested that hyperglycemia activates the Notch2 signaling pathway, which was negatively correlated with ALP expression levels. This inhibited osteo-blastogenesis [152]. Additionally, hyperglycemia has been shown to increase production of sclerostin, which induces adipogenesis by inhibiting WNT signaling in human BM-MSCs [153]. 

Some recent animal studies have shown higher bone marrow adiposity in diabetic models [151,154], which suggests the hypothesis that bone marrow fat composition may be a mechanism of diabetic fragility fractures [155,156]. In humans, one study measured a significantly higher bone marrow fat content in addition to predominant saturated lipid fraction in the diabetes mellitus group compared to healthy controls using proton magnetic resonance spectroscopy [157]. Similarly, another study demonstrated an alteration of bone marrow saturated to unsaturated fat composition using magnetic resonance imaging [29]. However, first, animal models are not consistently predicative of human responses [158], and, second, clinical studies showing increased bone marrow adiposity in diabetes mellitus have not ruled out obesity as a confounding factor. Patho-physiologically, T2DM is associated with insulin resistance. Therefore, cells from patients living with diabetes mellitus are less likely to accumulate lipids [159]. Increased bone marrow adiposity is known to correlate with altered levels of growth hormones, increased visceral adiposity, increased circulating lipids, and hypoleptinemia [28]. However, there is currently no evidence that suggests that diabetes mellitus directly accounts for increased bone marrow adiposity in humans. 

Recent investigations have shed light on impaired metabolic pathways in obesity, which results in chronic inflammation and insulin resistance. Therefore, this pre-disposes obese individuals to developing diabetes mellitus. White adipose tissue (WAT) in individuals living with diabetes mellitus has been shown to exhibit high levels of inflammation compared to WAT of obese individuals without diabetes mellitus [160]. Hypoxic conditions in adipose tissue caused by decreased perfusion of hypertrophic adipocytes leads to an upregulation of hypoxia-inducible factor 1-alpha (HIF-1α among other inflammatory genes [161,162]. Increased levels of inflammatory cytokines, in particular TNF-α, has been shown to induce insulin resistance [163,164]. Additionally, free fatty acids released by adipocytes produce ROS, which, in addition to hyperglycemia, exacerbates inhibited osteoblast proliferation and function maintained by a diabetic environment [165,166,167,168]. 

Thus, in vitro models have suggested that chronic inflammation in diabetes mellitus occurs as a result of a hyperglycemic bone marrow environment combined with oxidative stress, which inhibits the maturation of osteoblasts, and leads to a shift of MSC differentiation from osteo-blastogenesis to adipogenesis [136,169,170]. This leads to a vicious cycle of metabolic stress, which upholds a chronic inflammatory process that may de-mineralise trabecular bone [171], and result in the increased production of ROS, which has a direct impact on the differentiation and function of MSCs, osteoclasts, osteoblasts, and osteocytes [172]. In fact, the emerging understanding of T2DM as a cycle of chronic inflammation has opened windows to the development of anti-inflammatory treatment approaches [173].

A streptozotocin-induced T2DM diabetic mouse model showed evidence of suppressed expression of transcription factors required for the osteoblastic differentiation of MSCs in vitro [134]. This has been confirmed in a T2DM mouse model, where diabetic animals possessed fewer viable MSCs, which were functionally impaired ex vivo [174]. Exposing healthy cultured human MSCs to hyperglycemia, AGEs, and oxidative stress reduces the viable MSC population [54]. Thus far, only one study has been carried out to compare BM-MSCs isolated from individuals living with T1DM and healthy controls. This study suggested that BM-MSC cell count, cell morphology, and growth kinetics are not impaired despite long-term exposure to a diabetic stem cell environment in a young demographic [175]. However, to date, no studies have shown the effect of a diabetic environment on human MSCs isolated from individuals living with T2DM [176]. 

The sympathetic nervous system is responsible for mobilizing hematopoietic stem cells (HSCs) into the circulation, which have been shown to be inversely correlated with cardiovascular events in clinical studies [177,178]. It has been suggested that diabetes mellitus leads to remodeling and autonomic neuropathy of the bone marrow. Therefore, this affects the level of CD34+ cells in the blood [179]. These changes were averted in p66Shc knockout mice and are associated with the downregulation of the Sirt1 gene [180,181,182,183]. In a murine model, an insulin-resistant hyperglycemic environment leads to epigenetic changes in bone marrow via activation of JMJD3, a histone H3K27 demethylase, which leads to the increased expression of inflammatory cytokines. These changes persisted in peripheral monocytes, which leads to the hypothesis that epigenetic changes in the diabetic bone marrow environment leads to altered macrophage function and persistent wound inflammation [74]. Dipeptidyl peptidase-4 (DPP-4) inhibition has been shown to increase circulating HSCs in humans, which suggests that DPP-4 dysregulation plays a central role in diabetes mellitus-induced impaired HSC mobilization [184,185].

## 9. Effects of Insulin and Anti-Diabetic Drugs

Mice lacking an insulin receptor substrate, a mediator of insulin and IGF1 signaling, showed decreased bone formation and osteopenia due to reduced differentiation of osteoblasts [186,187], growth retardation, and a 60-fold higher expression of a hepatic IGF binding protein [188]. Additionally, osteoblasts lacking the insulin receptor substrate gene in an ex vivo model showed an upregulation of receptor activator of RANKL expression. Therefore, this stimulates osteo-clastogenesis in co-culture [186]. Conversely, a murine model of non-obese T2DM showed a reduced bone turnover rate, which was recovered by insulin treatment [189]. In humans living with T1DM, the incidence of osteoporosis or osteopenia was found to be significantly higher in patients before insulin treatment. After seven years of insulin treatment, bone turnover markers and BMD at all anatomical sites had significantly improved [190]. Although insulin is anabolic to bone and can restore markers of bone turnover and BMD, systematic review have identified no significant fracture reducing the potential for individuals living with diabetes mellitus on insulin treatment [191,192]. In fact, some epidemiological reports have shown an increased fracture risk in patients taking insulin, which may be secondary to an increased falls risk [192]. 

Metformin is routinely prescribed to patients as a first-line treatment T2DM, as recommended by consensus guidelines [193]. One population study has described metformin as having a potentially positive influence on fracture risk [191,194]. However, it is not clear whether this effect is secondary to blood sugar level optimisation or metformin directly interacting with progenitor cells to affect bone metabolism. In vitro studies examining the effect of metformin on MSCs have shown conflicting results. In rodent BM-MSCs, metformin stimulated osteoblastic activity and blocked adipogenesis [195]. Studies show decreased osteoclastogenesis in murine-derived preosteoclasts using supra-pharmacological concentrations of metformin [196,197,198]. However, some in vitro studies have shown MSC apoptosis following transplantation and decreased angiogenic potential of human MSCs treated with metformin [199,200]. In human-induced pluripotent MSCs, metformin enhanced osteoblastic activity by increasing ALP activity and mineralized nodule formation, which was partly mediated by the LKB1/AMPK pathway [201]. Bone turnover markers were measured following treatment with metformin in a clinical study [202,203], which showed decreased bone resorption (CTX-1) and a large decrease in bone formation (P1NP). However, this lacked a control arm [203]. 

After an initial response to metformin, many patients require additional anti-diabetic medications. Glitazones have detrimental effects on bone health and are, therefore, rarely prescribed [202,204]. The “incretin effect” (increased stimulation of insulin elicited by oral administration of glucose [205]) is proven to be significantly lower in diabetes mellitus compared to healthy subjects after a meal [206]. In murine models, the administration of the glucagon-like peptide 1 (GLP1), which is a hormone that facilitates the ‘incretin effect,’ has been shown to increase bone formation markers [207] and prevent the deterioration of the bone micro-architecture [208]. In vitro studies have shown GLP1 stimulates the proliferation of human MSCs and inhibits their differentiation into adipocytes [209] through GLP1 receptors expressed on progenitor cells [209,210]. 

GLP1 receptor analogues (GLP1RAs) are increasingly used because they aid weight loss and do not pose a risk of hypoglycemia [211]. One clinical study showed that the serum markers of calcium homeostasis (ALP, calcium, and phosphate) remained unaffected by exenatide treatment [212]. Additionally, a recent meta-analysis found no significant relationship between the use of GLP1RAs and fracture risk in T2DM in humans [213]. DPP-4 inhibitors are the second class of anti-diabetic drugs, which are designed to increase GLP1 levels. Recent reports have highlighted the impact of DPP-4 on circulating progenitor cells, which potentially ameliorates cardiovascular risk by facilitating HSC mobilization [185,214,215]. Nonetheless, thus far, meta-analysis has not established a cardiovascular benefit using DPP-4 inhibitors in patients [216]. Further translational research is required to thoroughly investigate the discrepancy between pre-clinical and clinical results.

In contrast, there is a strong evidence suggesting that treatment with sodium glucose cotransporter- 2 (SGLT-2) inhibitors positively affects cardiovascular and renal outcome in patients with T2DM [217,218,219]. Therefore, it has been hypothesized that this protective effect is caused by the increased mobilization of pro-vascular progenitor cells in bone marrow [220]. In one clinical trial, circulating CD133+ progenitor cells and monocytes with an anti-inflammatory phenotype were significantly raised and pro-inflammatory granulocyte precursors were significantly decreased following six months of treatment with empagliflozin [220]. A similar study measuring the effect of dapafliflozin showed an increase of CD34+KDR+ endothelial progenitor cells, which concurred with improvement in HbA1c, whereas circulating stem cells remained stable. This implies that the cardiovascular benefit may not directly involve circulating progenitor cells [221]. Despite these important advances, the mechanism of the cardiovascular and renal benefit of SGLT-2 inhibitors is still unknown. Furthermore, the epigenetic impact of these novel drugs on diabetes mellitus-induced bone fracture risk remains unexplored [222].

## 10. Conclusions

Recent literature shows that the fracture risk in diabetes mellitus increased more significantly than can be explained by changes in BMD and confounding factors, such as risk of falls [19,23]. Rather than influencing the mineral phase (BMD), it is thought that a diabetic environment primarily affects biomechanical properties of the bone by deteriorating its organic composition and bone material strength [29,30,33]. This occurs either directly through altered cross-link formation or indirectly through changes of cellular activity in osteoblasts and bone progenitor cells [41,42,50,223,224]. Besides altering gene expression and activity of osteoblasts [41,42], the diabetic environment significantly reduces the MSC population and viability [151,171]. In obese individuals living with T2DM, increased bone marrow fattiness may exacerbate MSC and osteoblast impairment by the release of cytokines and free fatty acids from hypoxic adipose tissue, which upholds a vicious cycle of chronic inflammation and inhibited osteoblastic activity (Figure 1) [165,168]. The combination of these changes eventually affects tensile strength and post-yield properties of the bone, which makes bone tissue in diabetes mellitus more vulnerable to microdamage accumulation, fragility fractures at most skeletal sites, and impaired fracture healing [32,225]. Decreased MSC population and impaired differentiation capacity may be the common link between impaired bone micro-architecture and higher incidence of non-union in patients living with diabetes mellitus [137,225]. Additionally, since vascularisation is mediated by MSCs [143,144], the reduced population and potential of progenitor cells may create vascular deficiencies in the fracture site, which can further impair diabetic fracture healing. A return to glucose homeostasis does not restore the capacity of previously diabetic MSCs, which reflects evidence outlining hyperglycemic memory in cells previously exposed to a diabetic milieu [64,65,66,67,68,69]. Therefore, it would be interesting to see studies investigating diabetes mellitus-induced epigenetic changes in precursor cells contributing to diabetic osteopathy. 

This review highlights the importance of efficient clinical management of patients suffering from diabetes mellitus, since adequately controlled diabetes mellitus has been consistently implicated to have a positive effect on bone health, which reverses bone impairments in some studies [130,189,190,208,226,227,228,229,230]. It is important to bear in mind that patients who are on a treatment regime causing hypoglycemic episodes are at a greater risk of sustaining fractures [231,232,233]. In clinical practice, health care professionals should focus on bone protection interventions and fall prevention strategies targeting patients at high risk of fracture [234]. Conventional risk assessment tools for osteoporosis such as BMD measurements and the FRAX score are not valid for predicting fracture risk in individuals living with diabetes mellitus [120,121,235]. Therefore, there continues to be a dire need for the investigation of novel methods of risk assessment, which possibly includes measurements of bone turnover and levels of AGEs, which can adjust for the altered metabolic state of diabetes mellitus [236,237]. MiRNAs are promising novel serum biomarkers, which could be used to identify individuals living with diabetes mellitus at a high risk of fragility fractures within the coming years [97,98]. Recent scientific developments in the understanding of the molecular pathways involved in diabetes mellitus have opened opportunities in new anti-inflammatory treatment approaches [173]. Further investigation is needed to clarify the mechanism of action through which diabetes mellitus affects the viability and differentiation capacity of the progenitor cell population, which will support translational research in the prevention of fragility fractures in patients suffering from diabetes mellitus in the future.

## Figures and Tables

**Figure 1 ijms-20-04873-f001:**
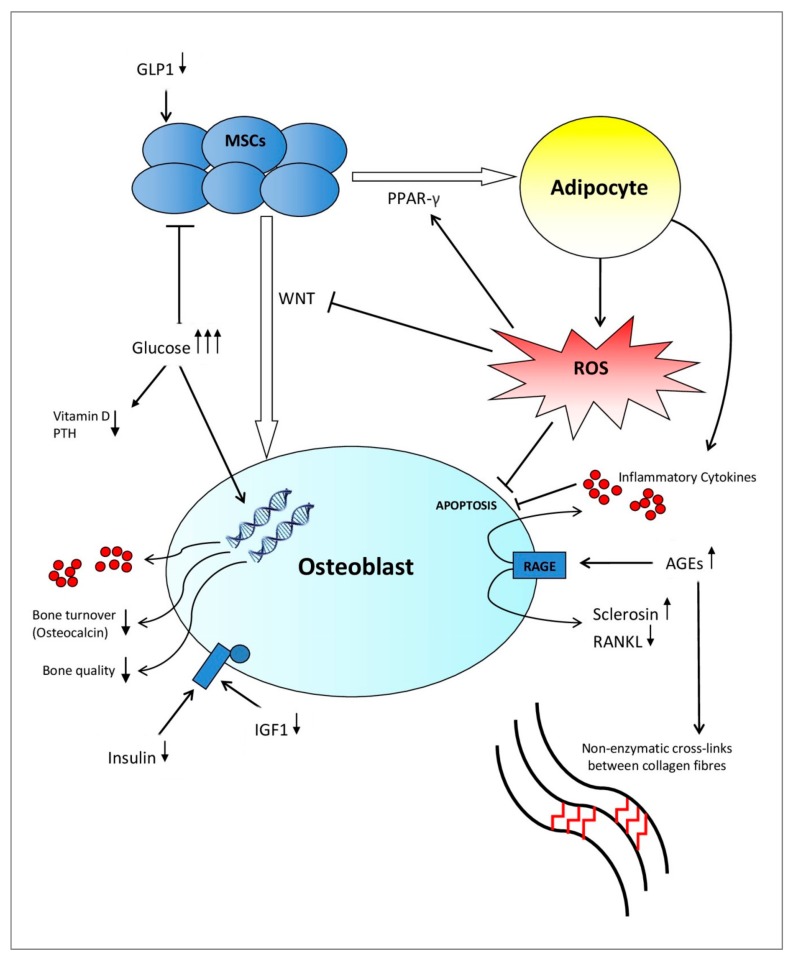
The interaction between osteoblasts, adipocytes, MSCs, and the marrow environment is altered in diabetes mellitus. Hyperglycemia directly alters gene expression associated with osteoblast activity by the inhibition of MSC maturation and metabolism, and indirectly alters bone metabolism by tampering with the PTH and Vitamin D system. Insulinopenia and low levels of IGF-1 exert an additional inhibitory effect on osteoblasts at different stages of diabetes mellitus. Increased production of adipocytes feed the cycle of chronic inflammation by producing ROS and inflammatory cytokines, which induce osteoblast apoptosis. ROS upholds this process by facilitating MSC differentiation into adipocytes by mediating PPAR-*γ* and reducing WNT transcription. Additionally, increased production of AGEs leads to non-enzymatic cross-links between collagen fibers and increased inflammation by the activation of RAGE. The accumulation of these patho-mechanisms ultimately leads to decreased bone quality and bone turnover in diabetes mellitus.

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
