# Peer review of "Impact of Diabetes Mellitus on Bone Health"

_ijms, 2019, doi:10.3390/ijms20194873_

Round 1

Reviewer 1 Report

The manuscript by Murray and Coleman examines an extremely important subject. The risk of fracture in patients with diabetes is, among the many diabetic complications, often underestimated by clinicians. The review is well written and I have few suggestions to improve the clarity of the text and its scientific impact:

The difference between bone mineral density in T1DM and T2DM should be moved in a dedicated paragraph and not in the introduction. Here authors should just describe the clinical/scientific impact of their topic and provide a short overview of the review structure. The figure1 as presented is pointless. The description in the legend is too brief and unclear. Each reported element and its interaction should be described in details, especially if reported throughout the text. To improve the quality of the manuscript, author should provide a concise overview about the effects of glucose lowering agents, especially with regards to those of the new generation such as GLP-1agonists and SGLT inhibitors, on bone health. Moreover, is there evidence of hyperglycemic memory in bone tissue?

Reviewer 2 Report

This is very interesting paper. I like this investigation.

Reviewer 3 Report

This is a well written paper aiming to explain molecular events associated with high risk of fracture among diabetic patients. The subject of work is very timely, since both incidence of fracture and diabetes constantly increases. I think that following issues can be interested to the readers and increase the quality of paper:

There is no methodology section. Please provide the criteria of references selection. Please add a paragraph or section presenting information how epigenetic mechanisms - induced by diabetes mellitus may be involved in high fracture risk among diabetic patients It is also very interesting how different hypoglycemic drugs affect fracture risk, metformin and insulin therapy are of special importance , since they are the most frequently used hypoglycemic drugs. 

Round 2

Reviewer 1 Report

Just a note, line 332: JMJD3 is not a transcription factor but a histone H3K27 demethylase, it is an enzyme.

Author Response

We have made the suggested edit in the attached draft of the manuscript. Please refer to the red text in line 332.